# PVT1 is a stress-responsive lncRNA that drives ovarian cancer metastasis and chemoresistance

Kevin Tabury[1,2,3,*], Mehri Monavarian[4,*], Eduardo Listik[4], Abigail K Shelton[5], Alex Seok Choi[4], Roel Quintens[2], Rebecca C Arend[6], Nadine Hempel[7,8], C Ryan Miller[5], Balázs Győrffy[9], Karthikeyan Mythreye[3,4]

**Metastatic growth of ovarian cancer cells into the peritoneal cavity requires adaptation to various cellular stress factors to facilitate cell survival and growth. Here, we demonstrate the role of PVT1, one such stress induced long non-coding RNA, in ovarian cancer growth and metastasis. PVT1 is an amplified and over-expressed lncRNA in ovarian cancer with strong predictive value for survival and response to targeted therapeutics. We find that expression of PVT1 is regulated by tumor cells in response to cellular stress, particularly loss of cell–cell contacts and changes in matrix rigidity occurring in a YAP1-dependent manner. Induction of PVT1 promotes tumor cell survival, growth, and migration. Conversely, reducing PVT1 levels robustly abrogates metastatic behavior and tumor cell dissemination in cell lines and syngeneic transplantation models in vivo. We find that reducing PVT1 causes widespread changes in the transcriptome leading to alterations in cellular stress response and metabolic pathways including doxorubicin metabolism, which impacts chemosensitivity. Together, these findings implicate PVT1 as a promising therapeutic target to suppress metastasis and chemoresistance in ovarian cancer.**

## Introduction

A feature of successful metastasis is the ability of cancer cells to adapt and survive under cellular stress. Mechanisms to do so include increased genomic instability, transcriptional and epigenetic changes, and acquisition of mutations (Hanahan & Weinberg, 2011). Such changes are coupled with signaling pathways and gene expression alterations that facilitate cancer cell survival and thereby tumor progression. Cellular stressors impacting most cancers include changes in oxygen tension (hypoxia), changes in ECM composition and matrix rigidity, alterations in cell–cell contacts and changes in anchorage-independent survival capabilities, oxidative, and metabolic stress as well as therapeutic treatments. Thus, defining stress dependent alterations is critical to our understanding of metastatic mechanisms.

Ovarian cancer is the fifth leading cause of cancer deaths with the highest mortality among all gynecological cancers (Siegel et al, 2020). Yet, the full etiology and pathophysiology continues to be delineated (Reid et al, 2017). Epithelial ovarian cancer is the most common type of ovarian cancer and is classified into different subtypes (high-grade serous carcinoma [HGS], low-grade serous carcinoma, mucinous carcinoma, endometroid carcinoma, and clear cell carcinoma) (Lheureux et al, 2019) that are marked by different genome amplification and genetic instability (Karnezis et al, 2017). All subtypes are exposed to cellular stressors during metastasis that involve the ovaries, the omentum, and the peritoneum. The metastatic peritoneal spread involves cell-ECM detachment, loss of cell–cell contacts, epithelial–mesenchymal transition (EMT) and shedding of cells from the tumor, followed by anchorage independent survival, re-attachment to new locations and re-establishment of cell–cell contacts (Kumari et al, 2021). Primary tumors in the fallopian tube and ovaries and secondary peritoneal growths in the abdominal cavity can also be hypoxic (Klemba et al, 2020), potentially further driving metastasis and chemoresistance in a feed forward manner (Rankin et al, 2016; Klemba et al, 2020). In addition, several prior studies have reported changes to cellular stiffness during metastasis in ovarian cancer (Swaminathan et al, 2011; Xu et al, 2012). Similarly, changes to the ECM and thereby matrix rigidity can also directly impact EMT responses and metastasis (McKenzie et al, 2018).

The Human Genome Project launched the era of non-coding RNAs (ncRNAs) (Lander, 2011). In these ncRNAs, small ncRNAs (<200

[1]Department of Biomedical Engineering, University of South Carolina, Columbia, SC, USA    [2]Radiobiology Unit, Belgian Nuclear Research Centre, SCK CEN, Mol, Belgium    [3]Department of Chemistry and Biochemistry, University of South Carolina, Columbia, SC, USA    [4]Division of Molecular Cellular Pathology, Department of Pathology, O'Neal Comprehensive Cancer Center, University of Alabama Heersink School of Medicine, Birmingham, AL, USA    [5]Division of Neuropathology, Department of Pathology, O'Neal Comprehensive Cancer Center, Comprehensive Neuroscience Center, University of Alabama Heersink School of Medicine, Birmingham, AL, USA    [6]Department of Gynecology Oncology, University of Alabama Heersink School of Medicine, Birmingham, AL, USA    [7]Department of Medicine, Division of Hematology Oncology, University of Pittsburgh School of Medicine Pittsburgh, PA, USA    [8]Department of Pharmacology, and Obstetrics and Gynecology, College of Medicine, Pennsylvania State University, Hershey, PA, USA    [9]TTK Cancer Biomarker Research Group, Institute of Enzymology, and Semmelweis University Department of Bioinformatics and 2nd Department of Pediatrics, Budapest, Hungary

Correspondence: mythreye@uab.edu
*Kevin Tabury and Mehri Monavarian contributed equally to this work.

nucleotides) such as microRNAs, small interference RNAs and PIWI-interacting RNAs have been intensively investigated for many years (Rupaimoole & Slack, 2017; Ozata et al, 2019). However, long ncRNAs (lncRNAs) (>200 nucleotides) and the contexts of their significance are still poorly understood. In ovarian cancer, several lncRNAs including *MALAT1*, *HOTAIR*, and *H19* are pivotal players in response to cellular stressors particularly genotoxic stress, metabolic stress and hypoxia and are also associated with tumorigenesis, metastasis and chemoresistance (Oncul et al, 2019). *PVT1* is one such lncRNA located on chr8q24.21 that is expressed at low levels in normal tissues but is also designated as an oncogene because of its amplification/up-regulation status in multiple cancers (Guan et al, 2007; Tseng et al, 2014) acting as a potential competing endogenous RNA for miRNAs (Li et al, 2017; Wang et al, 2019). PVT1 has complex roles in cancers including ovarian. On one hand it has been reported to suppress cell growth (Liu et al, 2015), whereas in others it has been reported to promote growth (Yi et al, 2020). Given the significant genetic instability of ovarian cancers and the ability of lncRNAs to impact metastasis through multiple mechanisms (Salamini-Montemurri et al, 2020), a first step to precisely defining lncRNA expression outcomes is to delineate and establish contexts that regulate lncRNA expression and activity.

Here, we investigate in detail the contexts and mechanism of lncRNA PVT1 expression and metastatic activities particularly in response to various stress factors in ovarian cancer. Our findings uncover PVT1 as a YAP1 dependent stress-responsive lncRNA, which can be altered transiently to drive metastasis and chemoresistance. We also discover a novel contribution of PVT1 in the regulation of doxorubicin resistance in this process.

# Results

## Clinical significance of PVT1 in ovarian cancer

To evaluate in detail PVT1 expression and amplification in broad cancer types, we evaluated The Cancer Genome Atlas (TCGA) datasets using cBioportal (Cerami et al, 2012; Gao et al, 2013). Analysis of somatic focal copy number gain events identified by GISTIC (v2.0) for serous ovarian cancers identified chromosome loci 8q24.21 as exhibiting the highest copy number gains (Fig 1A). Notably, MYC is located on 8q24.21 (Fig 1B). PVT1 was most significantly co-expressed with MYC, (Fig 1C and D—Pearson factor of 0.70 [$P$ = 3.11 × $10^{-46}$]). Among all cancers within the TCGA Firehose Legacy studies, serous ovarian cancer exhibited the highest frequency of alterations in PVT1 (43%) (Fig 1E and Table 1). Because serous ovarian cancers are marked by genome amplification (The Cancer Genome Atlas Research Network et al, 2011), we evaluated the effect of PVT1 amplification on PVT1 expression, by comparing PVT1 amplification versus expression and found that amplification and expression were highly correlated in ovarian cancer patients (Pearson factor of 0.46 $P$ = 4.27 × $10^{-17}$, Fig 1F). Hence, to evaluate the clinical significance of PVT1 expression changes alone, we conducted Kaplan–Meier survival analysis using KM plotter (Gyorffy et al, 2012) in TCGA datasets that included all ovarian cancer patients with a best cutoff analysis (Nagy et al, 2018). Log-rank statistics were

used to calculate the $P$-value and Hazard Ratio (HR). We found that patients with higher PVT1 expression had shorter overall survival (Fig 1G) and significantly shorter progression-free survival (PFS) (Fig 1H). Effects of PVT1 expression on PFS were also grade (Fig S1A) and stage dependent (Fig 1I) with an increased HR for Grade 2 (HR: 2.753, $P$ = 0.0256) (Fig S1A) and cancer Stage 4 (HR = 2.374, $P$ = 0.007) (Fig 1I). Kaplan-Meier analysis also suggests that patients with high PVT1 expression may benefit from targeted therapy (HR: 0.3344, $P$ = 0.033) compared with other treatment strategies such as chemotherapy (Fig 1J). This suggests that PVT1 has predictive value for survival and may play an important role in ovarian cancer and treatment outcomes.

## PVT1 expression levels are altered in response to cell density changes

Given the significant amplification of *PVT1* in ovarian cancer and the apparent clinical impact of increased expression of PVT1 (Fig 1), we examined the effect of stressors, which may pertain to the metastatic trajectory of ovarian cancers and by altering PVT1 expression in different cell lines. We first evaluated PVT1 expression at baseline using primers spanning PVT1 transcript isoforms containing–exon 1–2, exon 2–3, and exon 6–7 (Fig S1B) in a panel of human ovarian cancer (OVCA) cells lines, normal immortalized fallopian tube epithelial cells (p211) and surface epithelial cells (IOSE80) (Fig 2A). Whereas the detectable RNA levels varied between the cell lines, SK-OV3 cells exhibited the most robust levels of detectable PVT1 across all three exons of PVT1 (Fig 2A) consistent with prior reports of high PVT1 levels in this cell line (Sun et al, 2018; Yang et al, 2018; Yi et al, 2020). Using SK-OV3s, we next subjected the cells to a panel of pertinent ovarian cancer stressors including hypoxia, growth under anchorage independence and changes in cell density. We find that exposure to acute hypoxia (0.2%) for 24 h led to increased PVT1 RNA levels with reproducible and significant changes across exon 6–7 (Fig 2B $P$ = 0.0069) in SK-OV3 cells. Similarly, growing cells under anchorage independence led to fluctuations in levels of PVT1 as compared with growth under attached conditions, but this comparison did not reach statistical significance (Fig 2C) in SK-OV3 cells. In contrast to the modest changes observed in response to the above-described stressors, we found reproducible and significant changes in PVT1 RNA levels upon changes in cell density (Fig 2D). Specifically, switching cells between high- and low-density growth revealed marked and significant differences in PVT1 RNA levels, with increases across all exons seen in low density conditions (Fig 2D $P$ < 0.0001). Cell–cell contact, or absence thereof (low density) was confirmed by phase contrast microscopy and actin staining by immunofluorescence (Figs 2D and S2A). The effect of changing cell density on PVT1 levels was recapitulated in other cell lines as well, including human high grade serous OV90 cells across one exon (Fig S2B) and across exons 2–3 in the mouse ID8 cell line both in the presence and absence of Trp53 (ID8 Trp53−/−), the most commonly altered gene in OVCA (Fig 2E and F) ($P$ = 0.0112 and 0.0044, respectively).

However, in a subset of ovarian cancer cell lines including OVCAR3 and OVCA420, cell density alterations failed to induce changes in PVT1 expression (data not shown). We speculated that several serous ovarian cancer cell lines grow in clusters in the presence of cell–cell contacts, unlike SK-OV3 cells which exhibit

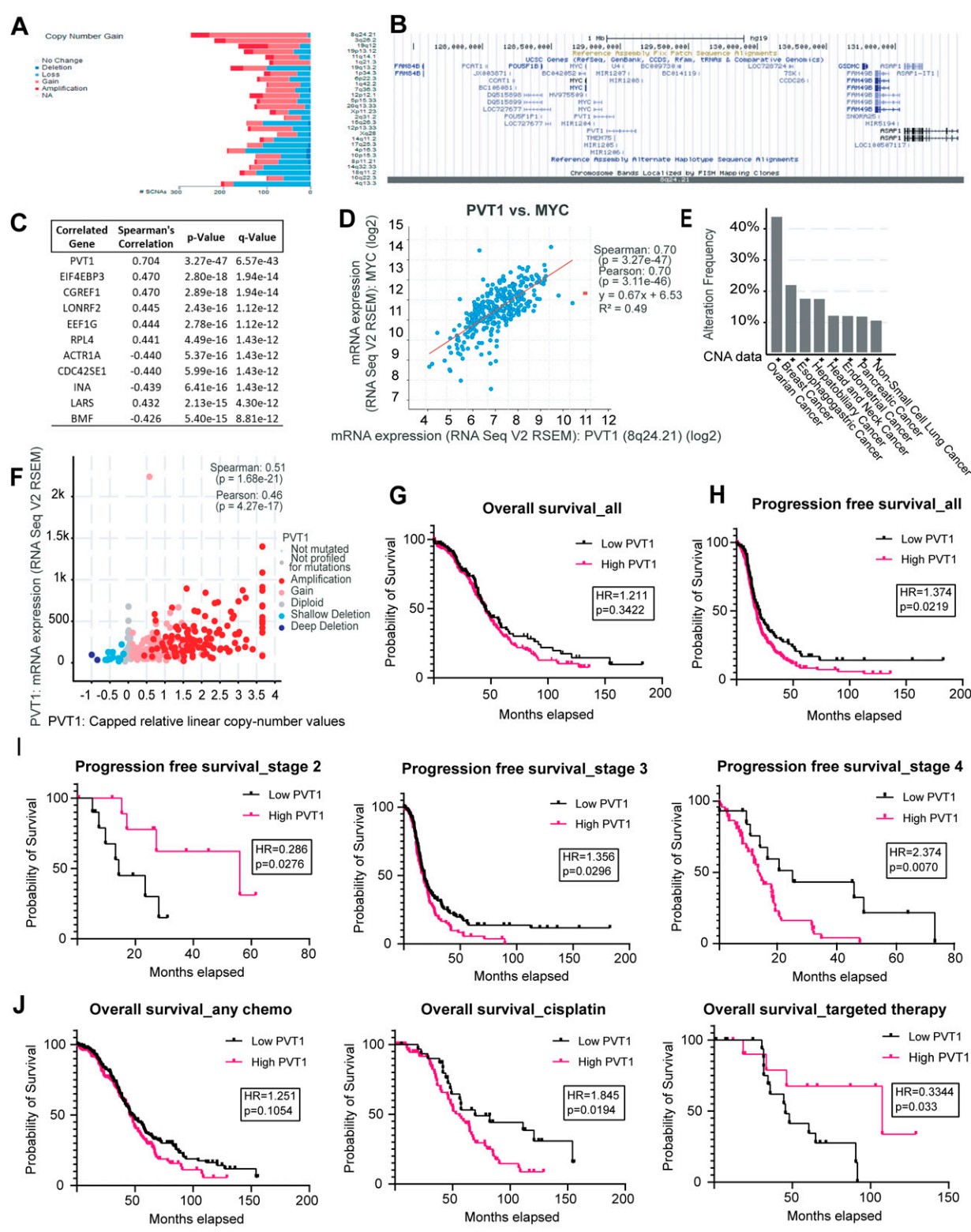

**Figure 1. PVT1 is amplified in ovarian cancer and impacts patient outcomes.**
(A) PVT1 Chromosomal loci copy number gain in ovarian cancer patients from TCGA. (B) Genomic view of chromosome loci 8q24.21. (C) Top 10 correlated genes with MYC (Spearman's correlation). (D) MYC versus PVT1 RNA co-expression in the TCGA ovarian cancer patient datasets (Pearson factor of 0.70 - $P = 3.11 \times 10^{-46}$). (E) PVT1 copy number alterations in cancers (top 8). (F) PVT1 RNA expression versus copy number alteration (Pearson factor of 0.46 - $P = 4.27 \times 10^{-17}$). (G) Kaplan–Meier analysis for overall survival (log-rank statistics) for ovarian cancer from TCGA. (H) Kaplan–Meier analysis for progression free survival (log-rank statistics) for ovarian cancer from TCGA. (I) Kaplan–Meier analysis for progression free survival by stage (log-rank statistics) for ovarian cancer from TCGA. (J) Kaplan–Meier analysis for overall survival by treatment (log-rank statistics) for ovarian cancer from TCGA (most patients with targeted therapy received bevacizumab).

**Table 1. Copy number alteration distribution from Fig 1E.**

| Cancer type | Alteration frequency | Alteration type | Alteration count |
|---|---|---|---|
| Ovarian Cancer | 0.345423143 | Homozygous deletion | 2 |
| Ovarian Cancer | 43.00518135 | Amplification | 249 |
| Breast Cancer | 0.186046512 | Homozygous deletion | 2 |
| Breast Cancer | 21.30232558 | Amplification | 229 |
| Esophagogastric Cancer | 0.32 | Homozygous deletion | 2 |
| Esophagogastric Cancer | 16.8 | Amplification | 105 |
| Hepatobiliary Cancer | 0.24691358 | Homozygous deletion | 1 |
| Hepatobiliary Cancer | 16.79012346 | Amplification | 68 |
| Head and Neck Cancer | 0.383141762 | Homozygous deletion | 2 |
| Head and Neck Cancer | 11.30268199 | Amplification | 59 |
| Endometrial Cancer | 0.168067227 | Homozygous deletion | 1 |
| Endometrial Cancer | 11.42857143 | Amplification | 68 |
| Pancreatic Cancer | 11.41304348 | Amplification | 21 |
| Non-Small Cell Lung Cancer | 10.12782694 | Amplification | 103 |

mesenchymal morphology and growth patterns (Yi et al, 2015). To test if PVT1 changes in response to cell density were a feature of mesenchymal cells, we induced EMT in epithelial ovarian cancer OVCAR3 and OVCA420 cells with TGF$\beta$-1. EMT was confirmed by increases in ZEB1 and SNAIL1 expression (Fig S2C). Post EMT mesenchymal cells indicated as OVCAR3-M and OVCA420-M, were then subjected to either high density or low-density growth conditions followed by evaluation of PVT1 levels. We find that PVT1 levels across all exons were increased under low density in both OVCAR3-M and OVCA420-M cells (Fig 2G and H) (OVCAR3: exon 1–2 $P$ = 0.002; exon 2–3 $P$ = 0.0013; exon 6–7 $P$ = 0.0134; OVCA420: exon 2–3 $P$ = 0.0229; exon 6–7 $P$ = 0.0086), mimicking the cell density response in SK-OV3 and ID8 cells (Fig 2D–F). We further expanded the cell density assay to DAOY (human medulloblastoma) and PC3 (human prostate cancer) cell lines and found an increase in PVT1 levels under low density (Fig S2D and E) in these non ovarian cell line models as well. These data indicate that isogenic mesenchymal cells are particularly sensitive to cell density associated PVT1 changes as compared with their epithelial counterparts. To next evaluate if PVT1 levels correlate with a mesenchymal gene signature in ovarian cancer patients we examined a panel of EMT genes and PVT1 in the ovarian cancer TCGA data. We find a positive correlation between PVT1 and EMT associated genes (Fig 2I) suggesting a strong correlation between post EMT regulation of PVT1 in ovarian cancer.

## PVT1 is depended on YAP1 in ovarian cancer

Cell density changes can lead to alterations in the cell cycle and concomitant changes to the Hippo pathway, a key modulator of cell survival in response to cellular stressors and a key negative regulator of YAP1 in various models (Piccolo et al, 2014; Dobrokhotov et al, 2018; Kim et al, 2019). Based on the cell density associated changes in PVT1 levels, we first evaluated if SK-OV3 cells under high density retain active Hippo signaling, as measured by

nuclear YAP1 localization (Dobrokhotov et al, 2018; Kim et al, 2019). We found nuclear accumulation of YAP1 in cells plated under low density, with significant exclusion under high density (Fig 3A and B $P$ < 0.0001). In addition, YAP1 target genes including CTGF and CYR61 were significantly up-regulated (CYR61 $P$ < 0.0001; CTGF $P$ = 0.0084) under low density compared with high density growth indicative of YAP1 function and activity under low density (Fig 3C).

To mechanistically extend the correlative observations of PVT1 RNA changes with YAP1 localization under low density, we tested if changing matrix rigidity would impact PVT1 RNA levels independent of the cell density associated changes. We first used fibronectin conjugated polyacrylamide hydrogels with an elastic modulus of 0.5 kPa (soft), 8 kPa (stiff), and normal plastic plates (in the order of GPa). We find that increasing matrix rigidity leads to increased PVT1 RNA levels (Fig 3D, 0.5 versus 8 kPa: exon 1–2 $P$ = 0.2523; exon 2–3 $P$ = 0.6295; exon 6–7 $P$ = 0.4684; 0.5 kPa versus 8 GPa: exon 1–2 $P$ < 0.0001; exon 2–3 $P$ < 0.0001; exon 6–7 $P$ < 0.0001; 8 kPa versus GPa: exon 1–2 $P$ = 0.0002; exon 2–3 $P$ < 0.0001; exon 6–7 $P$ = 0.0002). To confirm the direct correlation between YAP1 activity and substrate rigidity, CYR61 and CTGF mRNA expression was monitored and was found to be up-regulated with increasing matrix rigidity (Fig 3E). Serum is a known inhibitor of the Hippo pathway (Yu et al, 2012) and leads to increased YAP1 activity. To next test if depleting serum from low density cells when Hippo signaling is low and PVT1 is elevated, would lower PVT1 expression, we serum starved low density population of cells and evaluated PVT1 levels. We find that serum starved low density cells, further reduced their levels of PVT1 (Fig 3F) and at the same time also reduced YAP1 activity as indicated by the decrease of CYR61 and CTGF mRNA expression (Fig 3G). These data point to regulation of PVT1 co-incidentally with, or dependent on YAP1 activity.

To directly test if PVT1 expression depended on YAP1 levels, we used shRNAs' to lower YAP1 levels (Fig 3H) or inhibited the YAP1-TEAD association using the small molecule inhibitor verteporfin (Fig 3I) (Liu-Chittenden et al, 2012). We find that specific reduction of

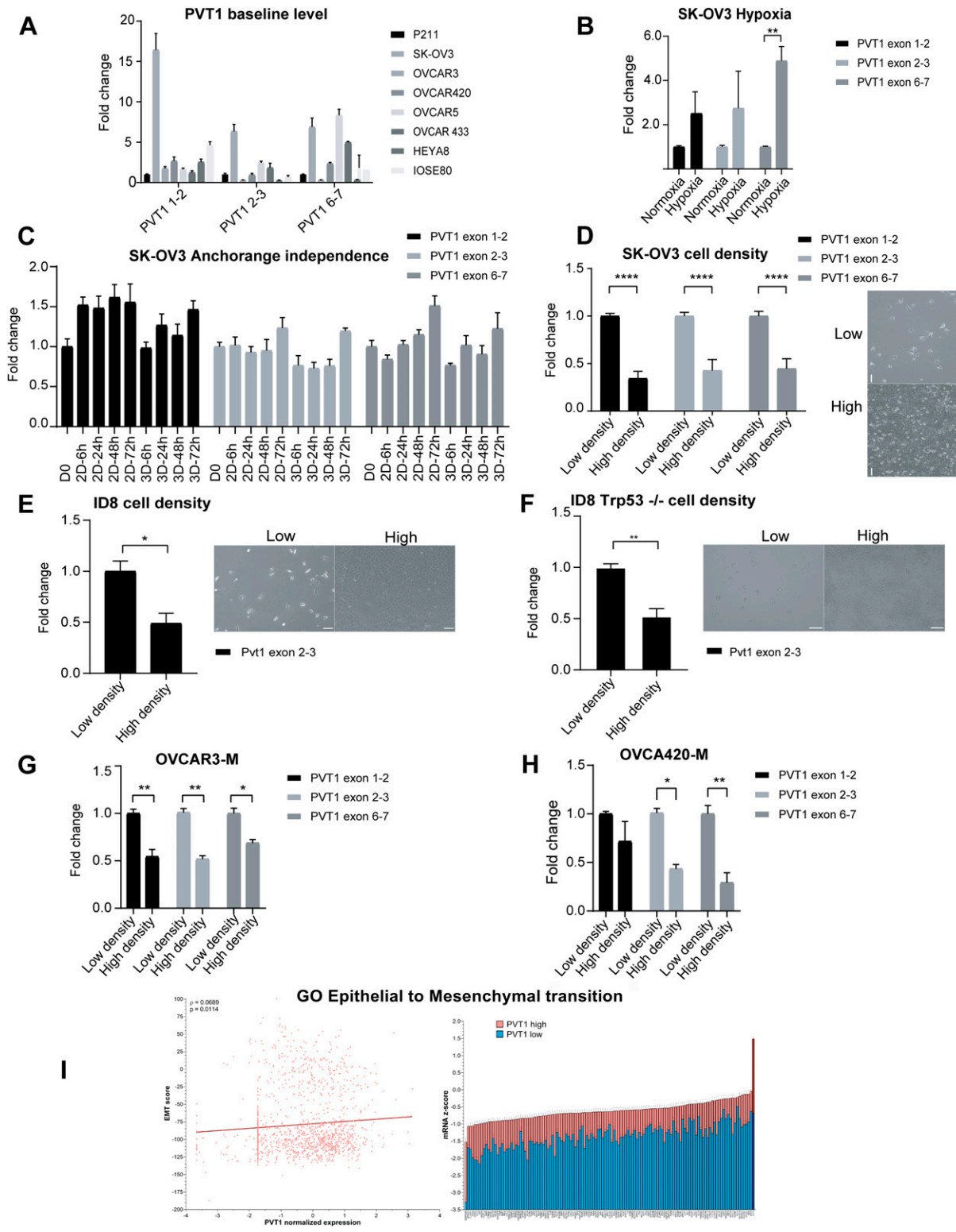

**Figure 2. PVT1 levels are altered in response to changes in cell density.**
**(A)** RT-qPCR analysis of PVT1 levels at baseline in a panel of human ovarian cancer and ovarian surface epithelial cell lines normalized to non-oncogenic fallopian tube epithelial cell line p211 (N = 2). **(B)** RT-qPCR analysis of PVT1 levels in indicated cells grown under hypoxia (0.2% Oxygen) normalized to the respective levels in normoxia conditions (two-way ANOVA-Sidak multiple comparisons test: n = 3; exon 1–2, P = 0.3760; exon 2–3, P = 0.4203; exon 6–7, P = 0.0069; N = 3). **(C)** RT-qPCR analysis of PVT1 RNA expression under anchorage independence (N = 2) normalized to D0 (day 0 levels). **(D)** RT-qPCR analysis of PVT1 RNA at different cell densities (inset figures) in SK-OV3 cells normalized to the low density levels (two-way ANOVA-Sidak multiple comparisons test: exon 1–2, P < 0.0001; exon 2–3, P < 0.0001; exon 6–7, P < 0.0001; N = 6).

YAP1 or the use of verteporfin significantly reduced PVT1 levels compared with control cells (Fig 3H and I). Based on this apparent dependency of PVT1 expression on YAP1 function, we next evaluated the clinical relevance of the PVT1-YAP1 relationship. We found a strong correlation between an inactive Hippo gene signature (where *YAP1* target gene signatures are expressed), and PVT1 expression in TCGA ovarian cancer datasets (Fig 3J). Together, these observations indicate a YAP1 dependency of PVT1 expression with functional correlations in ovarian cancer patients.

### PVT1 confers survival and pro-metastatic advantages to OVCA cells in vitro and in vivo

To test the effects of altering PVT1 levels on tumor cell behavior in vitro, we first used shRNAs (Fig 4A and B) to reduce PVT1 levels in both human SK-OV3 cells and mouse ID8 Trp53−/− cells. We found that lowering PVT1 resulted in reduced proliferation (MTT assay - *P* < 0.0001), migration potential of the tumor cells (transwell migration assay–*P* = 0.003) and clonogenic survival (*P* = 0.0088) in SK-OV3 (Fig 4C–E). Similar results were obtained upon reducing PVT1 levels in mouse ID8 Trp53−/− cells with shRNA, leading to significant reduction of proliferation (MTT assay—*P* < 0.0001), as well as migration potential (transwell migration assay–*P* = 0.0163) (Fig 4F and G). These observations were complemented with siRNAs targeting exon 4–5, exon 6, and 2 different sequences of exon 9 in mouse ID8 Trp53−/− cells (Fig S2F and G). Conversely, to evaluate the effect of increasing PVT1 expression, human *PVT1* was cloned into a pcDNA3.1(+) vector and expressed in SK-OV3 cells. Exogenous PVT1 expression was evaluated by RT-qPCR and FISH (Fig 4H and I). We found that overexpressing PVT1 resulted in increased proliferation (Fig 4Ji–*P* < 0.0001), increased migration (Fig 4Jii–*P* = 0.0007) and colony formation in a long term clonogenicity assay (Fig 4Jiii–*P* < 0.0001). These data suggest that although mouse and human PVT1 sequences vary between species (Noviello et al, 2018), both have likely conserved functions (Zampetaki et al, 2018) and promote tumorigenic behavior of cell lines.

Peritoneal growth and metastasis involves changes in cell–cell and cell-ECM contacts and survival of single cells that can attach to peritoneal organs (Cai et al, 2015; Kumari et al, 2021). We therefore tested if reducing PVT1 levels altered intraperitoneal tumor growth. We reduced Pvt1 expression in ID8 Trp53−/− cells with shRNAs targeting Pvt1 (shPvt1 targeting two sequences of exon 9) or non-targeting control (shctrl) (Fig 5Ai) and injected 5 × 10$^6$ viable cells within 96 h of Pvt1 knockdown into the peritoneal cavity of C57BL/6J mice (n = 12). Half the mice (n = 6) were euthanized after 4 wk to capture earlier effects of Pvt1 expression differences (mid-point of tumor growth as determined from pilot studies) and the remaining six mice were euthanized at 8 wk when they were moribund (Fig 5B).

At the 4-wk time point, we found that mice receiving control cells had already developed small omental and intraperitoneal lesions with observable tumor growth (Fig 5Aii and iii) compared with mice receiving shPvt1 cells (Fig 5A, black arrows). Omental weights were also measurably higher in mice receiving shctrl cells (Fig 5Aiv). The reduction in peritoneal lesions persisted over time, as shPvt1 cells had visibly (Fig 5Bi and ii) and measurably reduced overall tumor burden at 8 wk (Fig 5Biii, n = 6 for each group with one shctrl receiving mouse moribund after 8 wk). Assessment of the omentum, ovary, and peritoneal wall, all primary target tissues of metastatic ovarian cancer, revealed that cells expressing Pvt1 (shctrl) had seeded the peritoneum efficiently and invaded into the omental tissue (Fig 5Bi and ii). In contrast, lowering Pvt1 in shPvt1 cells led to significantly lower amounts of ascites fluid (Fig 5Biv *P* < 0.0001) and abdominal girth (Fig 5Bv *P* < 0.0001). These striking differences indicate that reducing Pvt1 leads to significantly less intraperitoneal tumor burden and demonstrates the critical role of PVT1 in ovarian cancer growth in vivo.

### PVT1 impacts global gene expression and regulates sensitivity to doxorubicin

Because PVT1 is a lncRNA with likely broad range effects, we evaluated the impact of PVT1 on global gene expression by performing RNA sequencing in human SK-OV3 cells upon silencing PVT1 using siRNAs (siPVT1). We found that 450 protein coding genes were differentially expressed between control (Ctrl vector) and siPVT1 cells with 50 additional genes found to be non-protein coding. The top 50 differentially expressed genes include both down-regulated and up-regulated genes in response to reducing PVT1 by siRNAs (Fig 6A). Lowered PVT1 levels were confirmed in the RNA-seq dataset (Fig S3A) as well as by RT-qPCR in biological replicates (Fig S3B). Furthermore, principal component analysis confirmed PVT1 siRNA status as a factor influencing the gene expression profiles (Fig S3C). The gProfiler web analysis tool was used to investigate all significant genes (n = 450). We find that PVT1 siRNA leads to substantial changes in the biological pathways associated with several stress responses and metabolism, specifically genes associated with doxorubicin metabolism (Fig 6B and C). Gene set enrichment analysis confirmed the "doxorubicin metabolism process" (Fig 6D) with genes such as *AKR1C1*, *AKR1C2*, and *AKR1B10* being significantly down-regulated in the siPVT1 samples (Fig 6C). Expression changes in these genes were also validated and confirmed by RT-qPCR for PVT1 knockdown (Fig 6E). In a reciprocal fashion, overexpression of PVT1 led to increased expression of *AKR1C1*, *AKR1C2*, and *AKR1B10* (Fig 6F). We next tested if siRNA to PVT1 sensitized SK-OV3 cells to doxorubicin. We found a 1.5-fold reduction in the IC$_{50}$ to doxorubicin in siPVT1 as compared with

---

**(E, F)** RT-qPCR analysis of Pvt1 RNA at different cell densities (inset figures) in indicated cells normalized to the low density levels (E) ID8 cells (unpaired two-tailed *t* test: exon 2–3, *P* = 0.0112; N = 3) (F) ID8 Trp53−/− cells (unpaired two-tailed *t* test: −exon 2–3, *P* = 0.0044; N = 3). **(G, H)** RT-qPCR analysis of PVT1 RNA in either low or high density of indicated post epithelial–mesenchymal transition cells after treatment with TGF-*β* for 96 h in (G) OVCAR3-M (two-way ANOVA-Sidak multiple comparisons test: exon 1–2, *P* = 0.002; exon 2–3, *P* = 0.0013; exon 6–7, *P* = 0.0134; N = 2) and (H) OVCA420-M (two-way ANOVA-Sidak's multiple comparisons test: −exon 2–3, *P* = 0.0229; exon 6–7, *P* = 0.0086; N = 2). **(I)** Correlation analysis of PVT1 RNA with gene ontology of epithelial to mesenchymal transition (z-score: value indicates the number of standard deviations away from the mean of mRNA expression in all the profiled samples). Error bars are indicated as SEM. Scale bar = 100 *μ*m. *P*-values are reported as ≥ 0.05 (ns), 0.01–0.05 (*), 0.001–0.01 (**), 0.0001–0.001 (***), and < 0.0001 (****).
Source data are available for this figure.

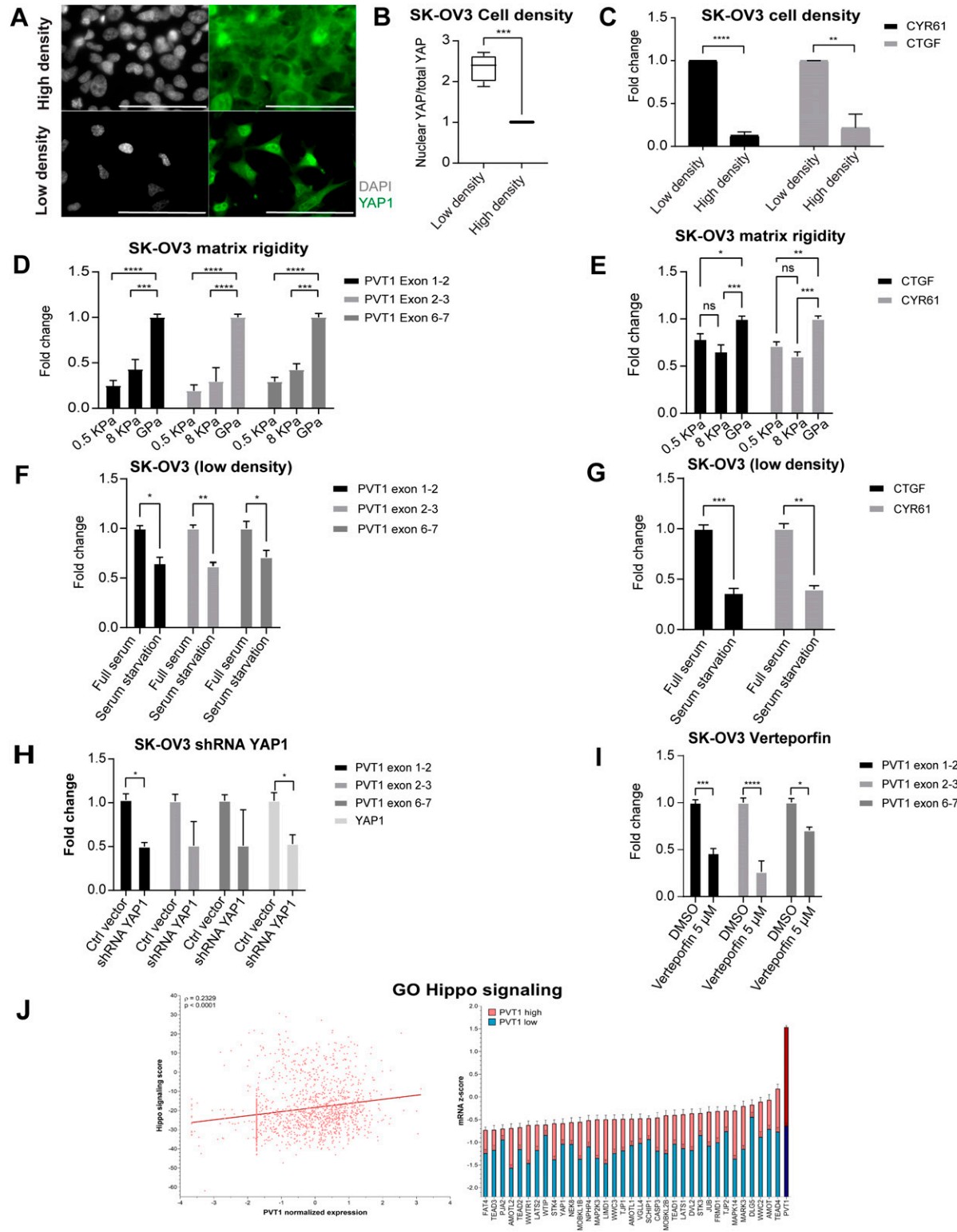

**Figure 3. PVT1 is regulated by YAP1.**
**(A)** Immunofluorescence of YAP1 under low and high density seeding in SK-OV3 cells (scale bar = 100 μm). **(B)** Quantification of the ratio of nuclear YAP to total YAP measured as an average intensity of total YAP in the nucleus/total YAP in the cell (unpaired two-tailed $t$ test: number of cells = minimum 70 cells per biological trial, $P$ = 0.0005; N = 3). **(C)** RT-qPCR of CTGF and CYR61 mRNA expression of cells grown in indicated cell densities normalized to levels in low density (unpaired two-tailed $t$ test: CYR61, $P$ < 0.0001; CTGF, $P$ = 0.0084; N = 3). **(D, E)** RT-qPCR of (D) PVT1 or (E) indicated genes in cells grown under different matrix rigidity conditions normalized to levels of normal plastic plates (GPa) (two-way ANOVA-Tukey multiple testing; N = 3). **(F)** RT-qPCR of PVT1 RNA from cells grown at low density and under either full serum or serum

siControl SK-OV3 cells (Fig 6G). Because human SK-OV3 cells do not express p53 (Yaginuma & Westphal, 1992), we evaluated if siPVT1 sensitized cells to doxorubicin even in the presence of p53 as the cell density dependent regulation of Pvt1 was largely p53-independent (Fig 2E and F). To test this, we used mouse ID8-IP2 cells that express wild type Trp53 (Nakayama et al, 2015) and evaluated doxorubicin sensitivity. We found that shRNA to Pvt1 significantly lowered doxorubicin $IC_{50}$ in these cells as well (Fig 6H) indicating conserved functions across species and likely independent of the p53 status. These observations indicate a novel correlation between PVT1 and doxorubicin resistance in ovarian cancer.

## Discussion

Here we demonstrate that *PVT1* is a contextual oncogenic lncRNA, amplified along with *MYC*, and a prognostic indicator in ovarian cancers that is dynamically altered in expression primarily in response to cellular stressors. Notably cell density and matrix stiffness changes, both of which converge on pathways associated with YAP1, lead to changes in PVT1. Our study is the first to report such a mechanism and suggest a potential feedforward relationship between YAP1 and PVT1.

We find a strong correlation between PVT1 and MYC in ovarian cancer, as has been previously documented (Tseng et al, 2014; Cho et al, 2018; Jin et al, 2019). Genome-wide association studies have identified chromosome 8q24.21 as a cancer risk locus in multiple cancers (Sud et al, 2017), but also specifically in ovarian cancer (Goode et al, 2010; Phelan et al, 2017). Besides harboring *MYC*, dysregulation and amplification of which is found in many human cancers (Dang, 2012), the 8q24.21 locus also harbors lncRNA *PVT1* 53 kb downstream of *MYC*. Bioinformatic analysis of TCGA datasets confirmed that chromosome 8q24.21 exhibits the highest copy number amplifications and that copy number alteration (CNA) frequency of *PVT1* is among the highest in ovarian cancer (Fig 1), highlighting the significance and relevance of *PVT1* to this cancer. Furthermore, PVT1 expression is also correlated with patient survival outcomes. Ovarian cancers are mostly detected at an advanced stage which is one of the causes of poor survival. Our findings are consistent with other reports of higher PVT1 mRNA expression in stage 3 compared with other stages (Chen et al, 2018; Ding et al, 2019). Interestingly, the impact of PVT1 expression differs for early-stage patients (stage II) where low PVT1 expression is correlated with poor PFS (Fig 1I). As ovarian cancers are marked by genome amplification, we investigated the correlation between PVT1 amplification and expression. Our data indicated a strong positive correlation between PVT1 amplification on chr 8q24.21

and expression in the TCGA cohort of primary patient tumors (Fig 1F).

We demonstrate here for the first time that PVT1 RNA expression is modulated by cell density, hypoxia, and matrix rigidity in both human and mouse ovarian cancer cell lines. Increases in PVT1 RNA expression under hypoxia have been previously described in non-ovarian cancers (Iden et al, 2016; Wang et al, 2018; Yu et al, 2020), but were somewhat modest in the ovarian cancer cell line SK-OV3 we tested here (Fig 2B). The most consistent and reproducible stressors that alter PVT1 expression are changes to cell density and mesenchymal status across cell lines, regardless of their baseline PVT1 levels (Fig 2A). The Hippo signaling pathway is altered in ovarian cancer and causes increased nuclear YAP1 and tumorigenesis (Zhang et al, 2011; Xia et al, 2014). Interestingly, modulation of PVT1 RNA expression after exposure to the stressors followed the RNA expression of YAP1 canonical target genes CTGF and CYR61 (Zhang et al, 2011; Piccolo et al, 2014), indicating a correlation between YAP1 nuclear localization and PVT1 RNA expression, which we also observed in patient data (Fig 3J). PVT1 expression levels were directly dependent on YAP1 levels as investigated through YAP1 knockdown as well as through disrupting YAP/TEAD interaction (Piccolo et al, 2014; Han, 2019). In the context of EMT, tumors often exhibit a spectrum of EMT states. Indeed, cell lines such as OVCAR3 and OVCA420 exhibit a more epithelial phenotype, whereas SK-OV3s exhibit a more intermediate to mesenchymal phenotype (Huang et al, 2013) and were found to be responsive to cell density dependent changes in PVT1. YAP1 plays a dual role as both inducer and effector of EMT (Zanconato et al, 2016) and has been shown to translocate from the cytoplasm to the nucleus after EMT induction through TGFβ treatment (Grannas et al, 2015). Consistently, we observed a significant sensitivity of EMT induced cells to cell density dependent induction of PVT1 as seen in post EMT OVCAR3 and OVCA420 cells that were rendered sensitive to cell density dependent PVT1 changes (Fig 2). Whereas it is likely that the PVT1 promoter possesses a TEAD-binding site (Verduci et al, 2017), the complex of YAP-TEAD on PVT1's promoter has not yet been reported and may involve indirect regulatory mechanisms as well. Little is known about modulators of PVT1 expression itself. Prior studies have demonstrated MYC (Tseng et al, 2014), FOXM1 (Xu et al, 2017), STAT3 (Zhao et al, 2018), and TP53 (Olivero et al, 2020) as regulators that can bind to PVT1's promoter. For instance, it has been shown that FOXM1, which binds to PVT1's promoter to induce PVT1 RNA expression (Xu et al, 2017), is a downstream effector of YAP1 (Fan et al, 2015). Thus, whether regulation is occurring here, directly, or indirectly at the promoter remains to be determined. A surprising observation is the lack of a role of p53

starvation as indicated, normalized to levels under full serum (two-way ANOVA-Sidak's multiple testing; −N = 2; exon 1–2, *P* = 0.0102; exon 2–3, *P* = 0.0068; exon 6–7, *P* = 0.0251). **(G)** RT-qPCR of indicated YAP1 target genes (CTGF and CYR61) of cells grown at low density and under either full serum or serum starvation as indicated, normalized to levels under full serum (two-way ANOVA-Sidak multiple testing: *CTGF*, *P* = 0.0009; *CYR61*, *P* = 0.0011; N = 2). **(H)** RT-qPCR of PVT1 RNA in shRNA YAP1 or Ctrl vector cells normalized to levels in Ctrl vector cells (two-way ANOVA-Sidak multiple testing: exon 1–2, *P* = 0.0199; YAP1, *P* = 0.0258; N = 2). **(I)** RT-qPCR of PVT1 RNA expression in cells treated with 5 *µ*M of Verteporfin or DMSO control normalized to levels in DMSO-treated cells (two-way ANOVA-Sidak multiple testing: exon 1–2, *P* < 0.0001; exon 2–3, *P* = 0.0002; exon 6–7, *P* = 0.0153; N = 3). **(J)** Correlation analysis of PVT1 expression with gene ontology of Hippo signaling pathway (z-score: value indicates the number of standard deviations away from the mean of mRNA expression in all profiled samples). Error bars are indicated as SEM. *P*-values are reported as ≥ 0.05 (ns), 0.01–0.05 (*), 0.001–0.01 (**), 0.0001–0.001 (***), and < 0.0001 (****).
Source data are available for this figure.

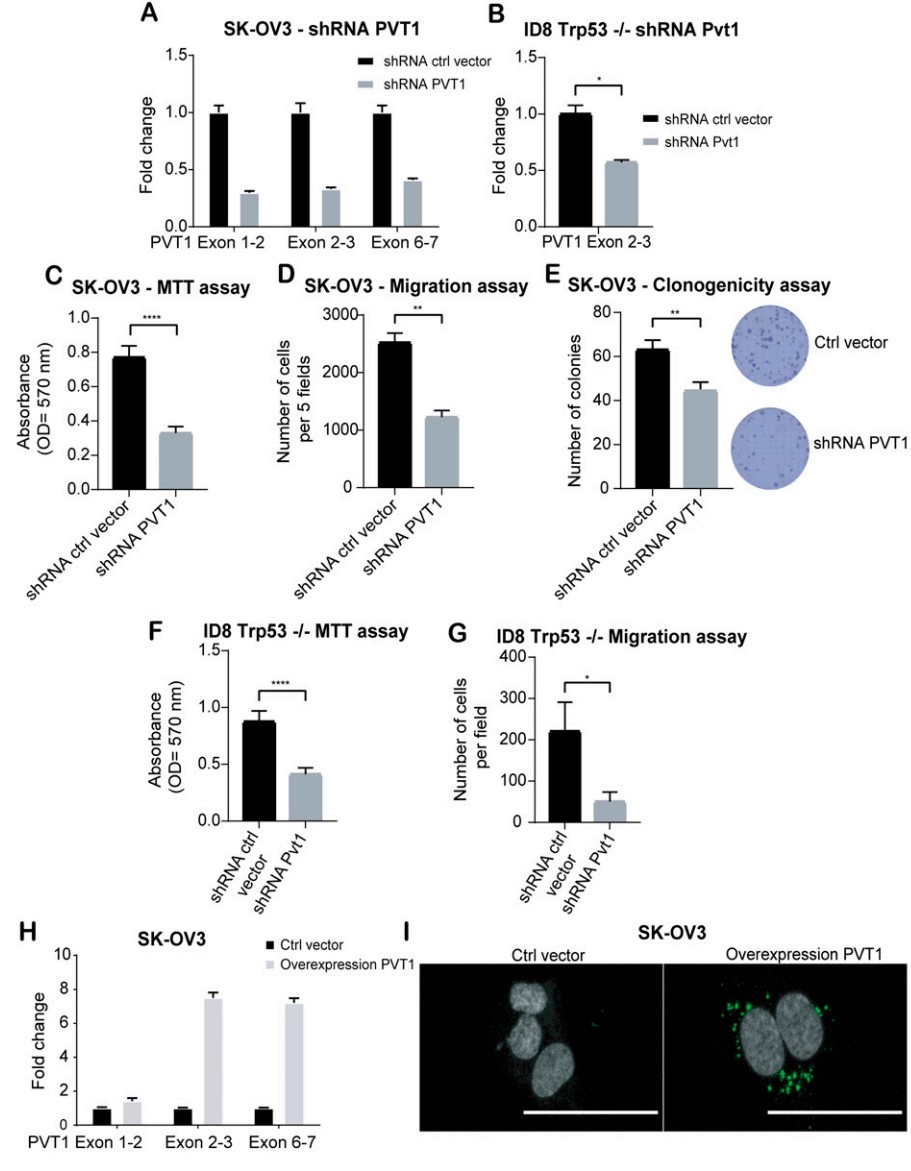

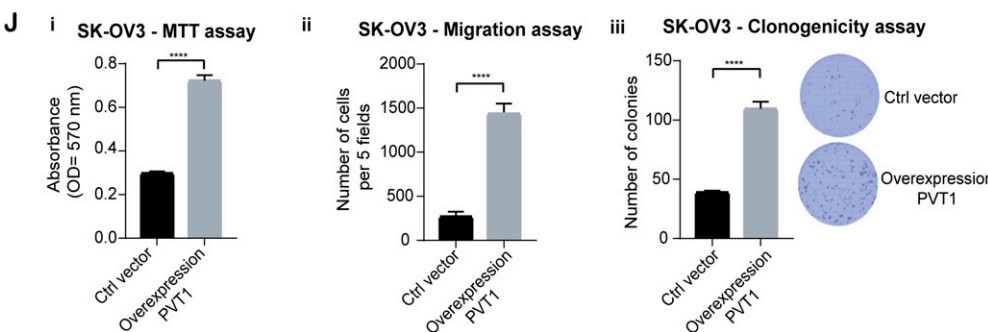

**Figure 4. PVT1 confers increased migration, and survival advantages to OVCA cells.**
**(A, B)** RT-qPCR of PVT1 in shPVT1 or shCtrl human SK-OV3 and mouse cells (ID8 Trp53−/−) (unpaired two-tailed t test: P = 0.0328) normalized to levels in shCtrl cells. **(C)** Absorbance values after completion of an MTT assay after 72 h of growth of shRNA PVT1 or shCtrl SK-OV3 cells (unpaired two-tailed t test: P < 0.0001; N = 12). **(D)** Transwell migration analysis of shRNA PVT1 or shCtrl SK-OV3 cells after 6 h of migration (unpaired two-tailed t test: P = 0.003; N = 3). **(E)** Numbers of colonies counted from a clonogenicity assay using shRNA PVT1 or shCtrl cells after 10 d (unpaired two-tailed t test: P = 0.0088; N = 6). **(F)** Absorbance values after completion of a MTT assay after 72 h using shRNA Pvt1 or shCtrl in mouse ID8 Trp53−/− cells (unpaired two-tailed t test: P < 0.0001; N = 12). **(G)** Transwell migration analysis of shRNA PVT1 or shCtrl mouse ID8 Trp53−/− cells after 6 h of migration (unpaired two-tailed t test: P = 0.0163; N = 3). **(H)** RT-qPCR of PVT1 in SK-OV3 cells expressing exogenous pcDNA3.1-PVT1 (overexpression PVT1) or ctl vector normalized to levels in control cells. **(I)** RNA-FISH images of cells expressing pcDNA 3.1 - PVT1 exogenously (overexpression PVT1) or ctl vector in SK-OV3 cells - scale bar 100 µm. **(J) (i)** Absorbance values after completion of a MTT assay after 72 h in indicated cells using PVT1 exogenously (overexpression PVT1) or ctl vector in SK-OV3 cells (Unpaired two-tailed t test: P < 0.0001; N = 12). **(ii)** Transwell migration analysis after 6 h of migration (unpaired two-tailed t test: P = 0.0007; N = 3) or **(iii)** Numbers of colonies counted from a clonogenicity assay in indicated cells (unpaired two-tailed t test: P < 0.0001; N = 6). P-values are reported as ≥0.05 (ns), 0.01–0.05 (*), 0.001–0.01 (**), 0.0001–0.001 (***), and < 0.0001 (****).
Source data are available for this figure.

in cell density dependent changes in Pvt1. Several prior studies have demonstrated PVT1 as a direct p53 target (Barsotti et al, 2012; Olivero et al, 2020). Our findings suggest that the YAP1 dependent mechanism of PVT1 may be p53 independent.

Functional assays with gain or loss of PVT1 RNA expression demonstrated PVT1's role in promoting survival and chemo-resistance in ovarian cancer cell lines and during intra peritoneal tumor growth, highlighting a pro-metastatic role for Pvt1 in ovarian

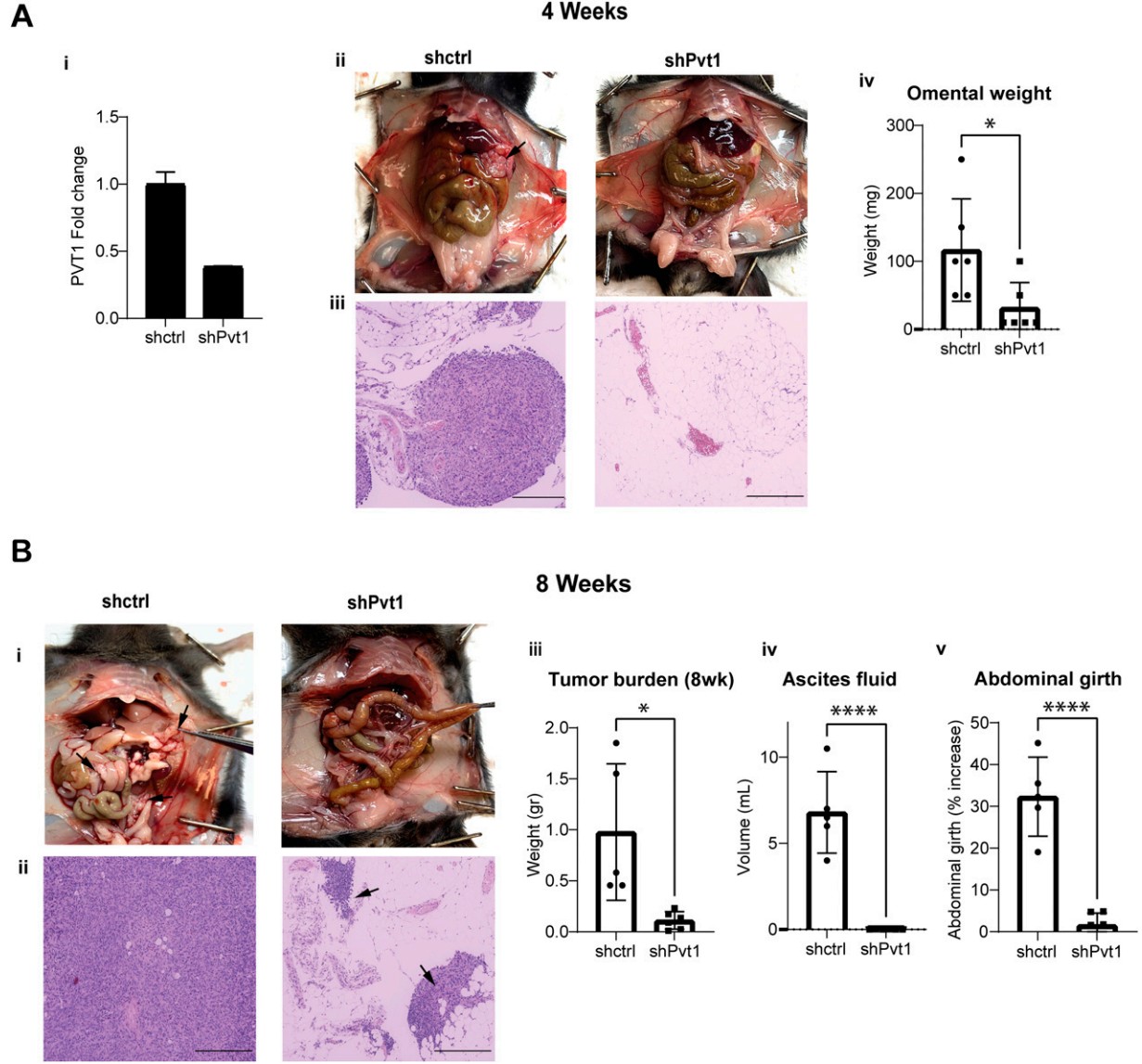

**Figure 5. Pvt1 promotes intraperitoneal tumor growth and metastasis in ovarian cancer.**
**(A) (i)** RT-qPCR of Pvt1 in shPvt1 or shCtrl mouse ID8 Trp53−/− cells normalized to levels in shctrl cells (N = 1). **(ii)** Representative mouse peritoneal images (above) and **(iii)** H&E-stained omental tissue sections after 4 wk following intraperitoneal injection of 5 × 10^6 shPvt1 or shCtrl mouse ID8 Trp53−/− cells. Black arrow in ii points to omental tumor/s. Scale bar = 275 μm. (N = 6 per group). **(iv)** Weight in grams of the omental tissue at 4 wk (unpaired two-tailed *t* test: *P* = 0.0325; N = 6). **(B) (i)** Representative mouse peritoneal images (above) and **(ii)** H&E-stained omental tissue sections after 8 wk after intraperitoneal injection of 5 × 10^6 shPvt1 or shCtrl mouse ID8 Trp53−/−-cells. **(iii)** Weight in grams of the total tumor burden after 8 wk (Unpaired two-tailed *t* test: *P* = 0.0113; N = 6). **(iv)** Volume in mL of ascites fluid collected from mice after 8 wk (unpaired two-tailed *t* test: *P* < 0.0001; N = 6). **(v)** Percent increase in abdominal girth of mice measured from day 40 to day 60. (unpaired two-tailed *t* test: *P* < 0.0001; N = 5 for shCtrl and n = 6 for shPvt1).
Source data are available for this figure.

cancer. Strikingly short-term knockdown of Pvt1 (within 96 h of shRNA administration) was sufficient to cause a significant reduction in tumor burden, suggesting that transient dynamic changes occurring during disease progression may be adequate at impacting disease outcomes. Our studies using the ID8Trp53−/− cells in immunocompetent models are also highly pertinent, as PVT1 was recently reported to be significantly correlated with CD8 T-cell infiltration in several cancer types (Li et al, 2020).

In the context of the models used here, transcriptomic analysis revealed a novel contribution of PVT1 in the regulation of aldo-keto reductases including AKR1C, AKR1C2, and AKR1B10. Aldo-keto reductases (AKR) have been demonstrated to regulate the metabolic processing of chemotherapeutic drugs such as doxorubicin, cisplatin, and paclitaxel, among others, and to cause chemoresistance in multiple cancers (Matsunaga et al, 2012; Zeng et al, 2017). *AKR1B10* has been linked to cellular resistance to doxorubicin by inducing the conversion of doxorubicin to doxorubicinol which is significantly

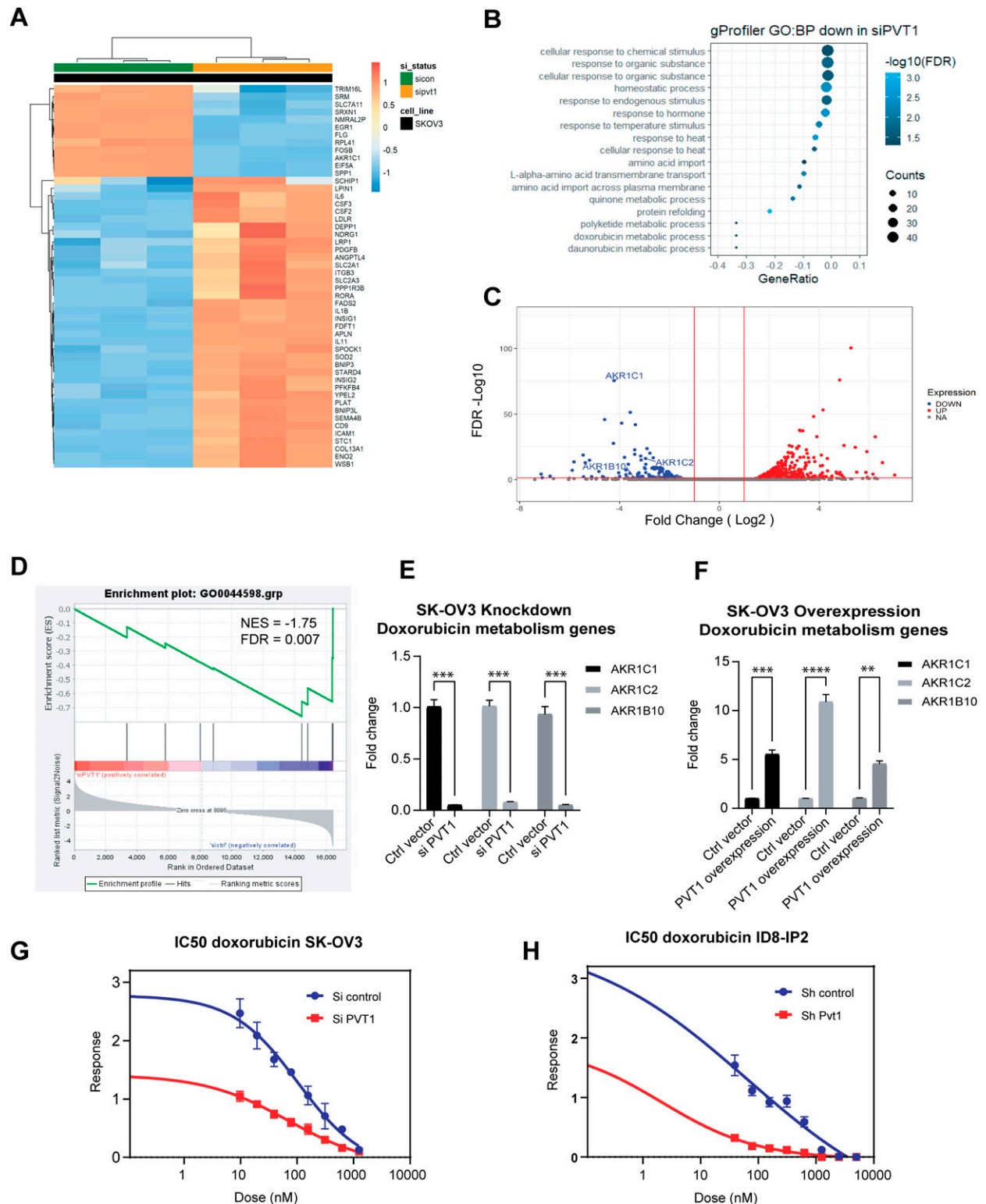

**Figure 6. RNA sequencing analysis reveals global transcriptional changes with specific changes to genes associated with doxorubicin resistance.**
**(A)** Heat map of the 50 top differentially expressed genes in siPVT1 versus Ctrl vector (sicon) in SK-OV3 cells. **(B)** gProfiler biological process enrichment analysis in siPVT1 SK-OV3 cells. **(C)** Volcano plot of doxorubicin resistance target genes (AKR1C1, AKR1C2, and AKR1B10) in siPVT1 SK-OV3 cells. **(D)** GSEA analysis of doxorubicin metabolic process enriched in siPVT1 SK-OV3 cells. **(E)** RT-qPCR of AKR1C1, AKR1C2, and AKR1B10 in siPVT1 and siCtrl cells (two-way ANOVA-Sidak multiple testing: $P < 0.0001$; N = 3). **(F)** RT-qPCR of AKR1C1, AKR1C2, and AKR1B10 in PVT1 overexpression and Ctrl vector SK-OV3 cells (two-way ANOVA-Sidak multiple testing: exon 1–2, $P = 0.0005$; exon 2–3, $P < 0.0001$; exon 6–7, $P = 0.0018$; N = 2). **(G)** IC50 values to doxorubicin (nM) in control and PVT1 knockdown in human SK-OV3. **(H)** IC50 values to doxorubicin (nM) in control and PVT1 knockdown in mouse ID8-IP2 cells. Source data are available for this figure.

less toxic (Heibein et al, 2012). *AKR1B10* has also been reported to have an oncogenic role across different cancer types (DiStefano & Davis, 2019). Indeed, lowering PVT1 increased the sensitivity to doxorubicin. Here again, the effect of Pvt1 on the sensitivity to doxorubicin was Trp53 independent. Therefore, targeting PVT1 to suppress the level of these AKR enzymes and consequently reducing inhibitory concentration of doxorubicin could pave the way toward minimizing the dose dependent cardiotoxicity of doxorubicin (Zhang et al, 2012). The exact mechanism by which PVT1 regulates AKRs expression remains to be elucidated.

In summary the broad impact of *PVT* on multiple pathways suggest that blocking PVT1 may be an attractive target to simultaneously suppress multiple pathways. Tools to silence lncRNAs are currently available (RNA interference, antisense oligonucleotides and genome editing [CRISPR/Cas9 system]), but none have been clinically used for PVT1 (Arun et al, 2018) and warrant further investigation for ovarian cancers.

# Materials and Methods

### Cell lines and reagents

Human ovarian cancer cell lines SK-OV3, PC3, DAOY, and OV-90-CRL-11732 were purchased from ATCC. OVCAR3 and OVCAR5 were obtained from NIH (NCI-60). OVCA420 and HeyA8 cells were a gift from Susan K Murphy. P211 were derived as described previously (Jazaeri et al, 2011; Varadaraj et al, 2015), IOSE80 cells were received from the Canadian tissue bank. Mouse ovarian surface epithelial cell line ID8 and ID8-IP2's were a kind gift from Jill K. Slack-Davis and ID8 Trp53−/− were a kind gift from Iain McNeish. SK-OV3, OVCAR3, OVCAR5, OVCAR433, OVCA420, and HeyA8 were cultured in RPMI-1640 (10-040-CV; Corning) supplemented with 10% FBS (35-010-CV; Corning) and 100 U/ml penicillin, 100 $\mu$g/ml streptomycin at 37°C in a humidified incubator containing 5% $CO_2$. PC3 were cultured in Ham's F-12K (21127022; Gibco), whereas DAOY were cultured in EMEM (670086; Gibco), both supplemented with 10% FBS at 37°C in a humidified incubator containing 5% $CO_2$. ID8 and ID8 Trp53−/− were cultured in DMEM (10-017-CV; Corning) supplemented with 4% FBS, 5 $\mu$g/ml insulin, 5 $\mu$g/ml transferrin, 5 ng/ml sodium selenite, and 100 U/ml penicillin, 100 $\mu$g/ml streptomycin at 37°C in a humidified incubator containing 5% $CO_2$. P211 and IOSE80 were cultured in DMEM supplemented with 10% FBS plus 100 U/ml penicillin, 100 $\mu$g/ml streptomycin. OV90 cells were cultured in a 1:1 mixture of MCDB 105 medium (M6395-1L; Sigma-Aldrich) and Medium 199 (M5017-10X1L; Sigma-Aldrich) containing 15% fetal bovine serum plus 100 U/ml penicillin and 100 $\mu$g/ml streptomycin. All cell lines were grown at 37°C in a humidified incubator containing 5% $CO_2$.

### *Antibodies*

Monoclonal mouse anti-YAP1 (sc-101199; Santa Cruz Biotechnology), Alexa Fluor 488 goat anti-mouse IgG, H+L (A11001; Invitrogen), and Alexa Fluor 488 Phalloidin (A12379; Invitrogen).

### *Reagents*

PolyHema (P3932; Sigma-Aldrich) was used to generate low attachment plates for the anchorage-independent assay; fibronectin (Recombinant Human Fibronectin, 8258-FN-050; R&D Systems); BSA (0903-5G; BioExpress); crystal violet (AA22866-14; VWR); PolyBrene (sc-134220; Santacruz Biotechnology); DMSO (BP231-100; Thermo Fisher Scientific); Paraformaldehyde (S898-07; Avantor); Triton-X100 (0694-1L; AMRESCO); absolute ethanol (Decon Laboratories); EMT was induced with 200 pM of Human recombinant TGF-$\beta$ 1 (240-B; R&D Systems); verteporfin (SML0534-5MG; Sigma-Aldrich); Paclitaxel (328420010; Thermo Fisher Scientific); Doxorubicin (BP25176-10; Fisher Bioreagent); and Cisplatin (HY-17394; Medchem Express).

### Bioinformatic analysis

Somatic focal copy number gain events identified by Genomic Identification of Significant Targets in Cancer (GISTIC) for TCGA ovarian serous cystadenocarcinoma were recovered from firebrowse (http://firebrowse.org/). Genome data viewer was used to investigate genes located in chromosome 8q24.21 (https://www.ncbi.nlm.nih.gov/genome/gdv/). TCGA data for ovarian serous cystadenocarcinoma (TCGA, Firehose Legacy, RNA Seq V2, 307 samples) was recovered from cBioportal (https://www.cbioportal.org/) and used to investigate mRNA expression, mRNA co-expression and CNAs. Survival data were generated from KM Plotter using the TCGA datasets, all patients and the best cutoff setting (Gyorffy et al, 2012; Nagy et al, 2018). Log-rank statistics were used to calculate the *P*-value (P) and HR.

Z-score normalized RNA-seq data from ovarian cancer patients were retrieved from TCGA datasets. Elements from the "GO Hippo signaling" and "GO EMT" gene sets (Mi et al, 2019) were filtered, and Hippo signaling score or EMT score was obtained from the direct sum of the gene's z-scores. Spearman rank correlation test was performed to assess whether PVT1 expression correlates to each obtained score. PVT1-high and -low groups of patients were also checked for mRNA expression of each gene within each gene set. Each group contained patients with the highest or lowest 10[th] percentile of PVT1 expression.

### Generation of stable overexpression/knockdown cells

For PVT1 and YAP1 knockdown, SK-OV3 and ID8 Trp53−/− cells were infected with 10 MOI of shRNAs or controls. All virus production was performed by the Functional Genomics Core of the Center for Targeted Therapeutics, University of South Carolina. In short, HEK293-FT cells, cultured in DMEM (high glucose) media with 10% FBS, 2 mM L-glutamine, penicillin (100 IU/ml), and streptomycin (100 mg/ml) were transiently transfected with specific lentiviral constructs and the packaging/envelope plasmids pMD2.G and psPAX2. The Virus-containing medium was collected at 48 and 72 h after transfection and centrifuged at 200*g* for 5 min at 4°C. The medium was filtered with 0.45-$\mu$m PES Syringe Filter and centrifuged at 21,000*g* for 16 h at 4°C. Pellets of lentivirus were re-dissolved in PBS at one-hundredth of the original volume. Lentiviruses were then diluted with fresh culture media to transduce target cells in the presence of 10 $\mu$g/ml PolyBrene (hexadimethrine bromide). Sequences for the multiple shRNAs are in Table 2. For the overexpression construct, the complete linear RNA PVT1 sequence (1,969 bp) was inserted in a pcDNA3.1(+) vector (SC1691 – GenScript). Control samples were achieved using an empty pcDNA3.1(+)

**Table 2. shRNA sequences.**

|  | shRNA – knockdown |
|---|---|
| Human PVT1 | Exon 2: GAGCTTCGTTCAAGTATTT |
|  | Exon 8: GAAATGTCCTCTCGCCTGC |
| Human control | SMARTvector Non-targeting hCMV-TurboRFP plasmid |
| Mouse pvt1 | Exon 9: CGAGTGTGAAGGAGCGAGT |
|  | Exon 9: TGACCTTATTGTAGACTAA |
| Mouse control | pHIV-Zsgreen plasmid |
| Human YAP1 | CCGGCCCAGTTAAATGTTCACCAATCTCGAGATTGGTGAACATTTAACTGGGTTTTTG |
|  | CCGGGACCAATAGCTCAGATCCTTTCTCGAGAAAGGATCTGAGCTATTGGTCTTTTTG |
| Human control | TRCN scr PLKO.1 plasmid |

plasmid. For YAP1, two shRNAs that include multiple transcript variants were inserted in a pLKO.1 vector (TRCN0000107265 and TRCN0000107268). Control samples were achieved using a TRCN scr PLKO.1 plasmid. The efficiency of knockdown or overexpression was performed by RT-qPCR (list of primers are in Table 3).

For transient PVT1 knockdown using siRNA, SK-OV3 cells were cultured to 50% confluency in six well plates. Human siRNA for PVT1 was achieved using a mixture of four siRNA (ACCUAUGAGCUUU-GAAUAA; GAGAACUGUCCUUACGUGA; CUUCAACCCAUUACGAUU and GUACGAACUUCAUCGCCCA) (smartpool R-029357-00-0005; Dharmacon). Control samples were achieved using Lincode Non-targeting Pool (D-001320-10-05; Dharmacon). 50 nM Pooled siRNA to human PVT1 or non-targeting siRNA from Dharmacon were used to transfect SK-OV3 cells for 48 h in full serum media carefully maintaining cell confluency to not exceed ~80%. Lipofectamine RNAiMAX Transfection Reagent (13778075; Thermo Fisher Scientific) was used to facilitate the transfection. This was followed by RNA extraction and verification of knockdown using primers to PVT1. Similarly, mouse siRNA for PVT1 was achieved using a mixture of four siRNA (AAGUAUACCCUUUAAGCGU; CGAGUGUGAAGGAGCGAGU; GAUGUCACACAGACGAUAA; and UGACCUUAUUGUAGACUAA) (smartpool R-065730-00-0005; Dharmacon).

## Functional assays

For Transwell migration assay, 8-$\mu$m-pore membranes (662638; Greiner Bio-One) were coated with 10 $\mu$g/ml fibronectin. A total of 20,000 cells were suspended in 100 $\mu$l serum-free medium and added to the upper chamber of each Transwell. The lower chamber was filled with 600 $\mu$l complete medium and incubated in a $CO_2$ incubator at 37°C for 6 h, after which fixation and staining occurred. For clonogenicity assay, 200 cells were cultured in each well of a six-well plate in complete medium for 10 d, and after cells were fixed and stained. Imaging was performed using EVOS M7000 inverted microscope (Thermo Fisher Scientific). Quantitation was performed by manual counting. For MTT assay, 1,000 cells were seeded in a 96-well plate for 24 h, then a final concentration of 1 mM MTT was added. The plates were incubated in a 37°C incubator for 2.5 h, and DMSO was added to dissolve the formazan crystals. Absorbance was measured using a Synergy HT plate reader at 570 nm.

## Cell density assay

Low cell density was achieved in seeding 35,000 cells per well in a six-well plate. High density was achieved in seeding 210,000 cells per well in a 24-well plate.

## Hypoxia assay

A hypoxia chamber was used to regulate the oxygen gas levels in the incubator to 0.2%. A total of 20,000 cells were seeded in 24-well plates for 24 h under hypoxia or normoxia.

## Matrix stiffness assay

Easy Coat hydrogels Softwell plates with 0.5 and 8 kPa were obtained from Matrigen (SW6-EC-0.5 EA, SW6-EC-8 EA) and regular six-well plates were coated with 10 $\mu$g/ml fibronectin before use. Cells were seeded at a density of 100,000 in a six-well plate and incubated for 24 h.

## IC50 determination

SK-OV3 and ID8-IP2 cells were cultured in 96-well plates at a density of 2,500 cells per well for 24 h. Then the medium was replaced with fresh medium containing Doxorubicin. After 72-h incubation with drugs, Sulforhodamine B (SRB, A14769-14; Alfa Aesar) assay was performed as previously described (Vichai & Kirtikara, 2006). IC50 was calculated using IC50 calculator in Graph Pad Prism.

## RNA fluorescence in situ hybridization and immunofluorescence

PVT1 visualization and localization were performed using a View-RNA cell plus assay probe for PVT1 (VA4-3082274-VCP; Thermo Fisher Scientific). The probe set per manufacturer covers region 423–1,392 and is designed to hybridize with human PVT1 specifically. Cells were seeded onto a NUNC eight-well chamber slide. DapB was used as a negative control, whereas GAPDH was used as a positive control. The procedure was performed as described by the manufacturer with no modifications. Images were acquired with a ZEISS LSM 800 confocal.

**Table 3. Primer sequences.**

| Gene ID | Primer sequence (5'→3') | Specie |
|---------|-------------------------|--------|
| RPL13a | Forw: AGATGGCGGAGGTGCAG | Human |
| | Rev: GGCCCAGCAGTACCTGTTTA | |
| PVT1 exon 1–2 | Forw: CACCTTCCAGTGGATTTCCTT | |
| | Rev: GACAGGCACAGCCATCTT | |
| PVT1 exon 2–3 | Forw: CTTCCTGGTGAAGCATCTGAT | |
| | Rev: TTCAGCCTCCACTTAAAGTACC | |
| PVT1 exon 6–7 | Forw: CTGTTTGCTTCTCCTGTTGC | |
| | Rev: GAACTCCTCAGCCTCCAAG | |
| CTGF | Forw: GCGTGTGCACCGCCAAAGAT | |
| | Rev: CAGGGCTGGGCAGACGAACG | |
| CYR61 | Forw: CGCCTTGTGAAAGAAACCCG | |
| | Rev: GGTTCGGGGGATTTCTTGGT | |
| YAP1 | Forw: TGACCCTCGTTTTGCCATGA | |
| | Rev: GTTGCTGCTGGTTGGAGTTG | |
| SNAIL 1 | Forw: AAGATGCACATCCGAAGCCA | |
| | Rev: CAGTGGGAGCAGGAGAATGG | |
| ZEB1 | Forw: CTGCTCCCTGTGCAGTTACA | |
| | Rev: GTGCACTTGAACTTGCGGTT | |
| AKR1C1 | Forw: GTCCTGGCCAAGAGCTACAA | |
| | Rev: CGCACATTTCTGTTTAGGCCAT | |
| AKR1C2 | Forw: ACGGAGTCATTGCCATTCAGA | |
| | Rev: CATGCAATGCCCTCCATGTTA | |
| AKR1B10 | Forw: AGAAACTGGAGGGCCTGTAA | |
| | Rev: CATGCAATGCCCTCCATGTTA | |
| Rpl13a | Forw: CAAGGTTGTTCGGCTGAAGC | Mouse |
| | Rev: GCTGTCACTGCCTGGTACTT | |
| Pvt1 exon 2–3 | Forw: CACTGAAAACAAGGACCGAAAC | |
| | Rev: ACAGACATTGGCAGTGGC | |

For immunofluorescence, cells were fixed in 4% paraformaldehyde for 15 min and permeabilized with 0.3% Triton X for 10 min at room temperature. Blocking was performed with 5% BSA, and cells were then incubated with a 1:100 dilution of anti-YAP1 overnight at 4°C. Then, the secondary antibody conjugated to Alexa 488 was used, at 1:200, for 1 h at room temperature in the dark. For actin, cells were stained for 20 min after fixation and permeabilization as above with Alexa 488 Phalloidin. Images were obtained using an EVOS M7000 microscope. Image J was used to perform the quantification.

### Animal studies

All mouse studies were performed in accordance with the Institutional Animal Care and Use Committee at the University of Alabama Birmingham. Female C57BL/6J mice were obtained from the Jackson laboratory. Five million ID8-Trp53−/− shCtrl and shPvt1 were injected intraperitoneally in mice (n = 12 per group). Mice were monitored daily with girth and weight measurement taken weekly. Animals were euthanized after either 4 wk (n = 6) or 8 wk (n = 6). At necropsy, ascites, if present, were collected and volumes measured, tumor weights in the omentum and other organs were recorded and collected when possible. For microscopic analysis of tissues, formalin-fixed tissues were processed, paraffin-embedded, and sectioned at 5 μm thickness and H&E stained at UAB's histology core.

### RNA-sequencing

Library preparation was performed on purified, extracted RNA using a KAPA mRNA HyperPrep Kit (Kapa; Biosystems) according to the manufacturer's protocol. High throughput sequencing with 75-bp single-end reads was performed on an Illumina NextSeq 550 using an Illumina NextSeq 500/550 High Output Kit. Reads were aligned to the human transcriptome GENCODE v35 (GRCh38.p13) using STAR and counted using Salmon (Dobin et al, 2013; Patro et al, 2017). Normalization and differential expression analysis were performed using the R package DESeq2 v1.34 (Love et al, 2014). Genes where there were fewer than three samples with normalized counts less than or equal to five were filtered out of the final data set. Benjamini–Hochberg–adjusted $P$-value of $P < 0.05$ and $\log_2$ fold change of 1 were the thresholds used to identify differentially expressed genes between treatment conditions. Pathway enrichment analysis was performed using GSEA (Mootha et al, 2003; Subramanian et al, 2005).

### Statistical analysis

Xenograft data were analyzed using parametric statistics. All real time PCR's are relative quantitative RT-PCR's (hereby referred to as RT-qPCR) and are a combined quantitation of an independent biological trials (indicated in legends) assayed in triplicate. All statistical analyses were conducted with GraphPad Prism Software and used tests referred to in the figure legends.

# Data Availability

The authors confirm that all data supporting the findings of this study are available within the article and supplementary data. RNA-seq data have been deposited in the NCBI-Gene Expression Omnibus (GEO) database under the accession ID GSE185933.

# Supplementary Information

# Acknowledgements

Funding for this work was provided partially by NIH R01CA230628 to K Mythreye and N Hempel. B Győrffy was supported by the FIEK_16-1-2016-0005 and 2020-4.1.1.-TKP2020 grants of the Ministry for Innovation and Technology

in Hungary. We also thank Antonis Kourtidis for generous guidance on the FISH methodology, Jill K Slack-Davis, Iain McNeish, Amir Jazaeri, Susan K Murphy, and the Canadian Ovarian Tissue Bank for the gift of cell lines.

## Author Contributions

K Tabury: conceptualization, resources, data curation, formal analysis, funding acquisition, validation, visualization, methodology, and writing—original draft, review, and editing.
M Monavarian: formal analysis, validation, investigation, visualization, methodology, and writing—review and editing.
E Listik: formal analysis and writing—review and editing.
AK Shelton: formal analysis, methodology, and writing—review and editing.
AS Choi: methodology and manuscript editing.
R Quintens: conceptualization, methodology, and writing—review and editing.
RC Arend: resources.
N Hempel: funding acquisition, methodology, and writing—review and editing.
CR Miller: software, validation, investigation, methodology, and writing—review and editing.
B Györrfy: formal analysis and writing—review and editing.
K Mythreye: conceptualization, formal analysis, supervision, funding acquisition, investigation, visualization, methodology, project administration, and writing—review and editing.

## Conflict of Interest Statement

The authors declare that they have no conflict of interest.

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
