## [Reviewer comments · Life Science Alliance]

Life Science Alliance

PVT1 is a stress responsive lncRNA that drives ovarian cancer metastasis and chemoresistance

Kevin Tabury, Mehri Monavarian, Eduardo Listik, Abigail Shelton, Alex Choi, Roel Quintens, Rebecca Arend, Nadine Hempel, C Miller, Balázs Györrfy, and Karthikeyan Mythreya

DOI: <https://doi.org/10.26508/lsa.202201370>

Corresponding author(s): *Karthikeyan Mythreya, University of Alabama at Birmingham*

Review Timeline:

Submission Date:	2022-01-10
Editorial Decision:	2022-02-08
Revision Received:	2022-05-17
Editorial Decision:	2022-06-10
Revision Received:	2022-06-24
Accepted:	2022-06-27

Scientific Editor: Novella Guidi

Transaction Report:

February 8, 2022

Re: Life Science Alliance manuscript #LSA-2022-01370

Dr. Karthikeyan Mythreye
University of Alabama at Birmingham
Pathology
320 B WTI
1824 6th avenue South
Birmingham, AL 35294

Dear Dr. Mythreye,

Thank you for submitting your manuscript entitled "PVT1, a YAP1 dependent stress responsive lncRNA drives ovarian cancer metastasis and chemoresistance" to Life Science Alliance. The manuscript was assessed by expert reviewers, whose comments are appended to this letter. We, thus, encourage you to submit a revised version of the manuscript back to LSA that responds to all of the reviewers' points.

Thank you for this interesting contribution to Life Science Alliance. We are looking forward to receiving your revised manuscript.

Sincerely,

B. MANUSCRIPT ORGANIZATION AND FORMATTING:

Reviewer #1 (Comments to the Authors (Required)):

The presented manuscript, Tabury et al., aims to determine the function and mechanisms of regulating long non-coding RNA PVT1 in ovarian cancer. The manuscript is well written and made on a very high technical level. The results advance the understanding of the progression of ovarian cancer.

The results indicated the Yap-dependent regulation of PVT1 expression and established the effect on ovarian tumor cells migration and proliferation. Further, the authors determined the effects on the PVT1 depletion by shRNA on an animal model of ovarian cancer and PVT1 dependent transcriptome. Unfortunately, the later experiments were somehow deficient since they were based on the effects of single shRNA. The results would be much more confident if authors will analyze the transcriptome of SK-O3 cells overexpressing PVT1 and determine the effects of the overexpression on intraperitoneal tumor growth and metastasis.

Minor points:

Fig 2. The cell-cell contact formation in dense vs. sparse cultures would be better represented by IF staining of contact proteins (e.g., N-Cadherin)

Fig 3B. The presentation of the Nuclear Yap/Total Yap ratio would more clearly represent the results.

Supplemental Figures 1C. The exome-specific difference in PVT1 expression required an explanation.

Reviewer #2 (Comments to the Authors (Required)):

Myhre and collaborators analyze the role and regulation of PVT1 in ovarian cancer cells. They discover that PVT1 expression is regulated by cell density and mechanical properties of the extracellular matrix. The loss of expression of PVT1 correlates with the loss of YAP1 activity. Coherently, silencing of YAP1 leads to downregulation of PVT1, suggesting a regulatory link between the two genes. This density-dependent regulation is particularly enhanced in mesenchymal ovarian cancer lines and can be triggered by EMT. Gain and loss of function studies suggest that PVT1 enhances aggressiveness and metastatic growth in ovarian cancer cells. In addition, the authors show that PVT1 expression modulates sensitivity to doxorubicin, suggesting that PVT1 overexpression may induce chemoresistance in cancer cells.

While others have already analyzed the expression and the role of PVT1 in ovarian cancer, the present work is original in the identification of regulatory signals modulating PVT1 overexpression and in proposing a role for PVT1 in chemoresistance. Limitations of the study are: (1) it is unclear whether cell density regulates PVT1 expression only in ovarian cancer cells or whether this is a general mechanism; (2) It is not clear whether PVT1 is a direct target of YAP1. For this reason, if the authors do not have additional evidence to support direct regulation, I strongly recommend softening the interpretation of the data and removing YAP1 from the title, which is misleading given the current content of the manuscript; (3) The data concerning chemoresistance is rather observational since there is little mechanistic insight which may tell us how PVT1 modulates resistance to doxorubicin.

Specific points:

1. Fig1E. Please report the frequency of mutations and CNA for each cancer type shown.

2. Fig1J. Please indicate the type of targeted therapy, either in the legend or in the text.

3. A limit is that the effect of hypoxia and anchorage independence on PVT1 expression was only performed with a single cell line: thus, results can't be generalized to other ovarian cancer lines. This should be clearly stated in the text.

4. Fig 2I and fig3J. Please describe in the legend what is the graph on the right (mRNA z-score).

5. Can the Author provide evidence that PVT1 is a direct target of YAP1?

6. Fig 6F: please show the IC50 plot and analyses in the supplementary figures: a bar plot reporting the IC50, as shown in figure 6F, is not informative.

Referee Cross-Comments

I agree with the points raised by reviewer one, especially concerning the limitation of the use of a single shRNA for the in vivo experiments.

Reviewer #1 (Comments to the Authors (Required)):

The presented manuscript, Tabury et al., aims to determine the function and mechanisms of regulating long non-coding RNA PVT1 in ovarian cancer. The manuscript is well written and made on a very high technical level. The results advance the understanding of the progression of ovarian cancer. The results indicated the Yap-dependent regulation of PVT1 expression and established the effect on ovarian tumor cells migration and proliferation. Further, the authors determined the effects on the PVT1 depletion by shRNA on an animal model of ovarian cancer and PVT1 dependent transcriptome.

Comment 1: Unfortunately, the later experiments were somehow deficient since they were based on the effects of single shRNA.

*We would like to thank the reviewer for allowing us to expand. We regret the lack of clarity in our explanation of methods. We have indeed used 2 shRNAs to Pvt1, both of which target exon 9 which seems to be sufficient to observe a drastic decrease in metastatic potential of the selected ovarian cancer cell line. We have now clarified this in **Results (line 238)** and **Methods (line 428)**. We would like to emphasize the significance of these findings as alternative splicing has been observed for lncRNA PVT1 (human and mouse) resulting in multiple splice variants with the mouse lncRNA Pvt1 having 3 transcript variants and the human lncRNA PVT1 only one. As the focus of a part of our studies was directed towards investigating the functional implications of altering PVT1 in ovarian cancer, with a future potential for therapeutic application, our findings are highly significant as targeting only exon 9 was highly effective.*

*We have however, also performed additional functional experiments using mouse siRNAs targeting exon 4-5, exon 6 and 2 different sequences of exon 9. We found that, similarly to our previous findings, invasion potential was decreased when Pvt1 levels were reduced. These data are now in **New supplementary Fig S2F-G**.*

Comment 2: The results would be much more confident if authors will analyze the transcriptome of SK-O3 cells overexpressing PVT1 and determine the effects of the overexpression on intraperitoneal tumor growth and metastasis.

*To address the impact of overexpression with respect to the transcriptomics, we performed RT-qPCR with SK-OV3 cells overexpressing PVT1 on the selected genes related to the doxorubicin pathway (AKR1C1; AKR1C2 and AKRB10) which indicated an upregulation of these genes when PVT1 is overexpressed. These data can now be found in **New Fig 6F** and suggest that reciprocal alterations of PVT1 are likely be reflected in in vitro transcriptomic studies*

*With respect to intraperitoneal growth studies using SKOV3, these studies could be interesting, however the ID8TP53-/- syngeneic model used here recapitulates human disease metastasis including into the omentum (PMID: 27530326) and opens up new avenues for future potential investigations into the role of Pvt1 as an immune related lncRNA where it has been recently implicated (PMID: 32081859). We have now discussed this on **lines 343-345** of the discussion.*

Comment 3: Fig 2. The cell-cell contact formation in dense vs. sparse cultures would be better represented by IF staining of contact proteins (e.g., N-Cadherin)

To be able to better visualize the cell-cell contacts as being the central issue raised by the reviewer, we have included phalloidin immunofluorescence staining of actin now in **New supplementary Fig S2A** for cell line SK-OV3 and ID8

Comment 4: Fig 3B. The presentation of the Nuclear Yap/Total Yap ratio would more clearly represent the results.

*We agree with the reviewer and modified it in **New Figure 3B** accordingly.*

Comment 5: Supplemental Figures 1C. The exome-specific difference in PVT1 expression required an explanation.

Ovarian and other cancer cells exhibit different degrees/extents of epithelial–mesenchymal transition (EMT). Based on prior reports, OV90 cells exhibit more of an epithelial subtype (like OVCAR3) rather than mesenchymal (PMID: 24201814). As observed with OVCAR3 and OVCAR420, induction of EMT sensitizes PVT1 expression based on cell density conditions. The degree of EMT (not evaluated in detail as part of these studies) could affect PVT1 expression. As alternative splicing exists for PVT1, we hypothesize that the EMT stage could be exon specific.

Reviewer #2 (Comments to the Authors (Required)):

Mythreye and collaborators analyze the role and regulation of PVT1 in ovarian cancer cells. They discover that PVT1 expression is regulated by cell density and mechanical properties of the extracellular matrix. The loss of expression of PVT1 correlates with the loss of YAP1 activity. Coherently, silencing of YAP1 leads to downregulation of PVT1, suggesting a regulatory link between the two genes. This density-dependent regulation is particularly enhanced in mesenchymal ovarian cancer lines and can be triggered by EMT. Gain and loss of function studies suggest that PVT1 enhances aggressiveness and metastatic growth in ovarian cancer cells. In addition, the authors show that PVT1 expression modulates sensitivity to doxorubicin, suggesting that PVT1 overexpression may induce chemoresistance in cancer cells. While others have already analyzed the expression and the role of PVT1 in ovarian cancer, the present work is original in the identification of regulatory signals modulating PVT1 overexpression and in proposing a role for PVT1 in chemoresistance.

Limitations of the study are:

Comment 1: it is unclear whether cell density regulates PVT1 expression only in ovarian cancer cells or whether this is a general mechanism.

*We thank the reviewer for this. We have now performed additional cell density experiments on human prostate cancer (PC3) and human medulloblastoma cells (DAOY) and found that similar to ovarian cancer cells, PVT1 is upregulated under low cell density. These data can now be found in **New Supplementary Figure S2D and E**.*

Comment 2: It is not clear whether PVT1 is a direct target of YAP1. For this reason, if the authors do not have additional evidence to support direct regulation, I strongly recommend

softening the interpretation of the data and removing YAP1 from the title, which is misleading given the current content of the manuscript.

*We agree with the reviewer that our data did not demonstrate that YAP1 interacts with PVT1 or its promoter region directly. We therefore specifically did not use the terminology “regulated by” to mislead the readers. We do however provide concrete evidence on the dependency of PVT1 expression on YAP1 levels using knockdown approaches (**Figure 3H**) and use of Verteporfin (**Figure 3I**) which blocks the complex of TEAD-YAP1 (Note: YAP1 has no direct DNA binding capability and cannot directly bind DNA. However, YAP1 can acts via TEADs). Additionally, our cell line and patient data converge on PVT1 expression dependency on YAP1 function (**Figure 3J**). We therefore believe that the use of “dependent” as terminology is adequate and highly appropriate. However, if both the editor and the reviewer strongly feel that the use of “dependent” is dramatically incorrect, we would agree to remove YAP1 from the title. We also modified the summary blurb (line 44), line 181 and 215 accordingly.*

Comment 3: The data concerning chemoresistance is rather observational since there is little mechanistic insight which may tell us how PVT1 modulates resistance to doxorubicin.

We modified line 283 accordingly. Line 359-360 also indicates that the mechanism remains to be elucidated.

Specific points:

Comment 4: Fig1E. Please report the frequency of mutations and CNA for each cancer type shown.

*We thank the reviewer for this request. Unfortunately, cBioportal only reports annotation of the mutation using Genome Nexus (which utilizes VEP with the canonical UniProt transcript). Consequently, no mutation data is available for PVT1 as PVT1 does not code for a protein. For clarity we discarded this information from the figure as it is not relevant. We also added **new Table 1** reporting the CNA for each cancer type shown.*

Comment 5: Fig1J. Please indicate the type of targeted therapy, either in the legend or in the text.

We thank the reviewer for this clarification. We adapted the legend of Fig1J with “majority of patients with targeted therapy received bevacizumab”.

Comment 6: A limit is that the effect of hypoxia and anchorage independence on PVT1 expression was only performed with a single cell line: thus, results can't be generalized to other ovarian cancer lines. This should be clearly stated in the text.

We agree with the reviewer and have modified lines stating specific cell lines: 147, 149 and 312 accordingly.

Comment 7: Fig 2I and fig3J. Please describe in the legend what is the graph on the right (mRNA z-score).

We thank the reviewer for this clarification. z-score: value indicates the number of standard deviations away from the mean of mRNA expression in the all profiled samples. This was added in the legend of Figure 2I and 3J.

Comment 8: Can the Author provide evidence that PVT1 is a direct target of YAP1?

YAP1 lacks DNA binding activity (PMID: 29366442) and can therefore not directly target PVT1's promoter. Typically YAP requires transcriptional co-activators, one of which is TEAD . As such we investigated the impact of Verteporfin (known to disrupt YAP-TEAD complex formation (PMID: 27621651). Please also see comment #2.

Comment 9: Fig 6F: please show the IC50 plot and analyses in the supplementary figures: a bar plot reporting the IC50, as shown in figure 6F, is not informative.

*We have now included the IC50 plots and analysis in **New Fig 6G** and **6H** accordingly.*

Referee

Cross-Comments

Comment 10: I agree with the points raised by reviewer one, especially concerning the limitation of the use of a single shRNA for the in vivo experiments.

This has now been addressed with reviewer 1 as well. Please see reviewer 1 – comment 1 or below:.

*“We would like to thank the reviewer for allowing us to expand. We regret the lack of clarity in our explanation of methods. We have indeed used 2 shRNAs to Pvt1, both of which target exon 9 which seems to be sufficient to observe a drastic decrease in metastatic potential of the selected ovarian cancer cell line. We have now clarified this in **Results (line 238)** and **Methods (line 428)**. We would like to emphasize the significance of these findings as alternative splicing has been observed for lncRNA PVT1 (human and mouse) resulting in multiple splice variants with the mouse lncRNA Pvt1 having 3 transcript variants and the human lncRNA PVT1 only one. As the focus of a part of our studies was directed towards investigating the functional implications of altering PVT1 in ovarian cancer, with a future potential for therapeutic application, our findings are highly significant as targeting only exon 9 was highly effective.*

*We have however, also performed additional functional experiments using mouse siRNAs targeting exon 4-5, exon 6 and 2 different sequences of exon 9. We found that, similarly to our previous findings, invasion potential was decreased when Pvt1 levels were reduced. These data are now in **New supplementary Fig S2F-G**.”*

June 10, 2022

RE: Life Science Alliance Manuscript #LSA-2022-01370R

Dr. Karthikeyan Mythreye
University of Alabama at Birmingham
Pathology
320 B WT1
1824 6th avenue South
Birmingham, AL 35294

Dear Dr. Mythreye,

Thank you for submitting your revised manuscript entitled "PVT1, a YAP1 dependent stress responsive lncRNA drives ovarian cancer metastasis and chemoresistance". We would be happy to publish your paper in Life Science Alliance pending final revisions necessary to meet our formatting guidelines.

- please address the final Reviewer 2 concern by removing YAP1 from the title
- please use the [10 author names, et al.] format in your references (i.e. limit the author names to the first 10)
- please double-check your figure callouts; you have a callout for Figure S1C, but this is not in the figure legend or the figure

Figure Check:

- Figure 2D needs scale bars and others could be more visible in 2E, 2F, Figure 3A

A. FINAL FILES:

B. MANUSCRIPT ORGANIZATION AND FORMATTING:

Sincerely,

Reviewer #1 (Comments to the Authors (Required)):

The quality of the manuscript significantly improved and all previous concerns are addressed. Therefore, the manuscript is acceptable for publication in its present form.

Reviewer #2 (Comments to the Authors (Required)):

I thank the Authors for taking the time to address my comments to their manuscript.
I only have a remark to the Authors rebuttal to my Comment 2, which I report below:

Comment 2: It is not clear whether PVT1 is a direct target of YAP1. For this reason, if the authors do not have additional evidence to support direct regulation, I strongly recommend softening the interpretation of the data and removing YAP1 from the title, which is misleading given the current content of the manuscript.

Authors' Rebuttal: We agree with the reviewer that our data did not demonstrate that YAP1 interacts with PVT1 or its promoter region directly. We therefore specifically did not use the terminology "regulated by" to mislead the readers. We do however provide concrete evidence on the dependency of PVT1 expression on YAP1 levels using knockdown approaches (Figure 3H) and use of Verteporfin (Figure 3I) which blocks the complex of TEAD-YAP1 (Note: YAP1 has no direct DNA binding capability and cannot directly bind DNA. However, YAP1 can act via TEADs). Additionally, our cell line and patient data converge on PVT1 expression dependency on YAP1 function (Figure 3J). We therefore believe that the use of "dependent" as terminology is adequate and highly appropriate. However, if both the editor and the reviewer strongly feel that the use of "dependent" is dramatically incorrect, we would agree to remove YAP1 from the title. We also modified the summary blurb (line 44), line 181 and 215 accordingly.

Reviewer 2 rebuttal: In all fairness, given that the YAP1-dependent expression of PVT1 was evaluated only in a single cell line and considering that in the manuscript there is no reported effort to try to understand how YAP1 mechanistically regulates PVT1 expression, then it follows that linking YAP1 activity to PVT1 expression is not the main focus of the manuscript. Thus, I still think that a title not mentioning YAP1 would better reflect the content of the manuscript.

Response to remaining issues

- please address the final Reviewer 2 concern by removing YAP1 from the title
- please use the [10 author names, et al.] format in your references (i.e. limit the author names to the first 10)
- please double-check your figure callouts; you have a callout for Figure S1C, but this is not in the figure legend or the figure

Figure Check:

- Figure 2D needs scale bars and others could be more visible in 2E, 2F, Figure 3A

These have all been corrected.

Reviewer #2 (Comments to the Authors (Required)):

I thank the Authors for taking the time to address my comments to their manuscript.

I only have a remark to the Authors rebuttal to my Comment 2, which I report below:

Comment 2: It is not clear whether PVT1 is a direct target of YAP1. For this reason, if the authors do not have additional evidence to support direct regulation, I strongly recommend softening the interpretation of the data and removing YAP1 from the title, which is misleading given the current content of the manuscript.

Authors' Rebuttal: We agree with the reviewer that our data did not demonstrate that YAP1 interacts with PVT1 or its promoter region directly. We therefore specifically did not use the terminology "regulated by" to mislead the readers. We do however provide concrete evidence on the dependency of PVT1 expression on YAP1 levels using knockdown approaches (Figure 3H) and use of Verteporfin (Figure 3I) which blocks the complex of TEAD-YAP1 (Note: YAP1 has no direct DNA binding capability and cannot directly bind DNA. However, YAP1 can act via TEADs). Additionally, our cell line and patient data converge on PVT1 expression dependency on YAP1 function (Figure 3J). We therefore believe that the use of "dependent" as terminology is adequate and highly appropriate. However, if both the editor and the reviewer strongly feel that the use of "dependent" is dramatically incorrect, we would agree to remove YAP1 from the title. We also modified the summary blurb (line 44), line 181 and 215 accordingly.

Reviewer 2 rebuttal: In all fairness, given that the YAP1-dependent expression of PVT1 was evaluated only in a single cell line and considering that in the manuscript there is no reported effort to try to understand how YAP1 mechanistically regulates PVT1 expression, then it follows that linking YAP1 activity to PVT1 expression is not the main focus of the manuscript. Thus, I still think that a title not mentioning YAP1 would better reflect the content of the manuscript.

*We have now changed the title to: **PVT1 is a stress responsive lncRNA that drives ovarian cancer metastasis and chemoresistance***

June 27, 2022

RE: Life Science Alliance Manuscript #LSA-2022-01370RR

Dr. Karthikeyan Mythreye
University of Alabama at Birmingham
Pathology
320 B WT1
1824 6th Avenue South
Birmingham, AL 35294

Dear Dr. Mythreye,

Thank you for submitting your Research Article entitled "PVT1 is a stress responsive lncRNA that drives ovarian cancer metastasis and chemoresistance". It is a pleasure to let you know that your manuscript is now accepted for publication in Life Science Alliance. Congratulations on this interesting work.

DISTRIBUTION OF MATERIALS:

Again, congratulations on a very nice paper. I hope you found the review process to be constructive and are pleased with how the manuscript was handled editorially. We look forward to future exciting submissions from your lab.

Sincerely,
